# Engineering Inhalable Therapeutic Particles: Conventional and Emerging Approaches

**DOI:** 10.3390/pharmaceutics15122706

**Published:** 2023-11-30

**Authors:** Aditi Negi, Shubham Nimbkar, Jeyan Arthur Moses

**Affiliations:** 1Computational Modeling and Nanoscale Processing Unit, National Institute of Food Technology, Entrepreneurship and Management—Thanjavur, Ministry of Food Processing Industries, Government of India, Thanjavur 613005, Tamil Nadu, India; 2Food Processing Business Incubation Centre, National Institute of Food Technology, Entrepreneurship and Management—Thanjavur, Ministry of Food Processing Industries, Government of India, Thanjavur 613005, Tamil Nadu, India

**Keywords:** dry powder inhaler, pulmonary delivery, aerosolization, particle engineering, therapeutic efficiency

## Abstract

Respirable particles are integral to effective inhalable therapeutic ingredient delivery, demanding precise engineering for optimal lung deposition and therapeutic efficacy. This review describes different physicochemical properties and their role in determining the aerodynamic performance and therapeutic efficacy of dry powder formulations. Furthermore, advances in top-down and bottom-up techniques in particle preparation, highlighting their roles in tailoring particle properties and optimizing therapeutic outcomes, are also presented. Practices adopted for particle engineering during the past 100 years indicate a significant transition in research and commercial interest in the strategies used, with several innovative concepts coming into play in the past decade. Accordingly, this article highlights futuristic particle engineering approaches such as electrospraying, inkjet printing, thin film freeze drying, and supercritical processes, including their prospects and associated challenges. With such technologies, it is possible to reshape inhaled therapeutic ingredient delivery, optimizing therapeutic benefits and improving the quality of life for patients with respiratory diseases and beyond.

## 1. Introduction

The delivery approach is a critical determinant of the efficacy and biological performance of therapeutic ingredients, both pharmaceuticals and nutraceuticals. Pulmonary delivery of therapeutic ingredients is recognized for its rapid, non-invasive nature and capability to avoid the first-pass metabolism. The lungs are characterized by a large surface area, abundant blood supply, and high permeability [1]. Both local deposition and systemic pulmonary delivery can serve two purposes, namely: local deposition of therapeutic ingredients, mainly for the treatment of respiratory ailments, and systemic absorption of molecules [2]. Given these, in many cases, pulmonary delivery can help achieve a better therapeutic impact as compared to parenteral and oral routes.

Typically, the different approaches for pulmonary delivery include propellant-driven metered dose inhalers, nebulizers, and dry powder inhalers. Propellant-driven metered dose inhalers, also termed pressurized metered dose inhalers (pMDIs), carry the therapeutic ingredient in the form of a liquid or suspension, which is subsequently converted into an aerosol with the assistance of propellant. In this context, the use of chlorofluorocarbon-based propellants has rendered this approach non-sustainable [3]. On the other hand, nebulization requires specific equipment that can convert the liquid formulations into aerosols using physical means such as high-velocity air, ultrasound, or high-frequency vibrating plates. These may be associated with high equipment costs and challenges with portability [4]. Dry powder inhalers (DPIs) store and deliver therapeutic ingredients in the form of powders, providing higher stability, easier handling, propellant-free delivery, and portability, making them widely popular. Therefore, DPIs have been used for the delivery of several therapeutic ingredients and therapeutic molecules, such as insulin [5], anti-tuberculosis therapeutic ingredients [6], asthma therapeutic ingredients, vaccines [7], proteins, and peptides [8].

The effective delivery of DPI formulations depends on the deposition site in the lungs, which in turn depends on the properties of particles, specifically the mass median aerodynamic diameter (MMAD). Typically, smaller particles with a MMAD in the range of ~3 µm have a better chance of effective deposition [9]. Additionally, the solubility of the powders, which is in turn linked with the crystallinity, also plays a significant role. Herein, approaches used for the development of such particles govern such properties and are commonly classified as top-down and bottom-up approaches. Top-down approaches involve a size reduction in bulk materials to develop particles; on the other hand, in bottom-up approaches, particles are built up from much smaller sizes. Typically, top-down approaches offer ease of operation, whereas bottom-up approaches offer better control over particle properties.

Careful optimization of formulations and precise control of process parameters are essential to developing particles possessing desired properties. Given these, this article focuses on different conventional and novel techniques for the development of DPI formulations. Furthermore, the properties of particles, engineering considerations, and their relation to fate upon delivery are emphasized.

## 2. Physicochemical Properties of Inhaled Therapeutic Ingredients

The pulmonary delivery of drugs can occur through two approaches, namely intranasal and oral inhalation. The oral inhalation approach is found to be more effective due to minimal concentration loss. The mechanisms involved in the deposition of inhaled dry powder are inertial impaction, gravitational sedimentation, and Brownian diffusion. Apart from these, interception and electrostatic precipitation also play a role in minor parts [10]. Larger particles (>5 µm) are removed by impaction in the upper respiratory tract, while a combination of impaction, gravitational sedimentation, and Brownian diffusion takes place in the mid and lower airways. Particles with a size smaller than 5 µm significantly enhance the chances of penetration and retention [11]. Once deposited at the desired site, the active pharmaceutical ingredients (API) carried by these particles is readily absorbed into systemic circulation through alveoli, offering a rapid onset of action. The physicochemical properties of inhaled powders are crucial for their effectiveness in pulmonary delivery (Figure 1). These properties can impact the dispersion, deposition, and overall performance of inhaled powders. Therefore, optimizing such properties is essential for the effective development of inhaled powder formulations, as they can significantly impact the delivery efficiency and therapeutic effectiveness of treating respiratory conditions. In this context, the key physicochemical properties of inhaled powders include particle size and distribution, particle morphology, crystallinity, hygroscopicity, and surface charge, as described below.

### 2.1. Particle Size and Distribution

The particle size and size distribution of dry powder aerosols for inhalation can impact molecule deposition in the lungs. Over the years, this aspect has been evaluated through various experimental approaches, such as the Anderson cascade impactor (ACI), multistage liquid impinger (MLI), and next-generation impactor (NGI). Also, the behavior of formulated powder is described through different metrics like the median diameter of the size distribution (Dv50), fine particle fraction (FPF), mass median aerodynamic diameter (MMAD), and geometrical standard deviation (GSD) [12]. 

FPF measures the fraction of particles within a specific size range that is capable of reaching a target site in the lungs. Typically, the FPF is calculated for particles in the respirable range (e.g., 1–5 μm) as a percentage of the total ED. MMAD represents the diameter of the particle in an aerosol that has half the aerosol mass below it; it is a critical parameter for determining the deposition site within the respiratory tract. Particles with smaller MMAD values can reach the deeper lung regions. Research findings have shown that particles having aerodynamic diameters in the order of 1–5 µm are most suitable for generating aerosols that are intended for inhalation applications [13]. The formation of aerosols can be restricted in cases involving high and low cohesiveness of the particles. Particle cohesiveness increases when the particle is smaller than 1 μm and decreases when it is larger than 5 μm. High and low particle cohesiveness may hinder aerosol formation; however, research shows that particles between 1–5 μm have optimum cohesiveness for aerosolization [10]. 

To illustrate the impact of particle size and distribution, Chew et al. conducted an experiment where they generated three distinct powders of mannitol with aerodynamic diameters measuring 2.3, 3.7, and 5.2 µm. These powders were then administered using two inhaler devices, namely the Rotahaler^®^ (Allen and Hanburys, Ware, UK) and Dinkihaler^®^ (Aventis, Paris, France) at airflow rates of 60 L/min and 120 L/min, to compare their performance [14]. Results indicated that particles with aerodynamic diameters of 2.7 µm showed 12% aerosolization (by weight) and those with 5 µm showed 22% aerosolization (by weight) when subjected to an air supply of 60 L/min. In the case of the Rotahaler^®^, when subjected to 120 L/min air flow, there was a notable increase in the aerosolization of particles with an MMAD of 2.7 µm, resulting in a weight increase to 25%. However, particles with an MMAD of 5 µm did not have a significant impact under the same conditions. Furthermore, in the case of the Dinkihaler^®^, particles with aerodynamic diameters of 2.7 µm and 5 µm recorded aerosolization rates of 63% and 32%, respectively, when subjected to an airflow rate of 60 L/min [14].

### 2.2. Particle Morphology

Particle morphology can influence the aerosolization and deposition behavior of inhaled powders [15]. Irregular or elongated particles may exhibit different behaviors than spherical particles. This is because irregular-shaped particles have a low area of contact, correspondingly lower Van Der Walls forces, and thus a lower tendency to form aggregates. Studies have reported different particle shapes, such as spherical, pollen-shaped, cube-shaped, plate-shaped, and needle-shaped, and their flowability, aerosolization, and deposition patterns. While spherical and pollen-shaped particles have larger aerodynamic diameters, they also have higher FPF values. Pollen-shaped particles have rough and porous surfaces, resulting in lower particle density. Low-density particles tend to be more easily carried away by the airstream and thus are less likely to be deposited in the lungs. The aerodynamic diameters are smaller than their actual size, and their irregular surfaces restrict close interactions and thus reduce the cohesive force responsible for dispersion [16,17]. For instance, in their work on the solid particles of bovine serum albumin, Chew and Chan observed that wrinkled surface particles (3.1 µm) dispersed better than smooth surfaces (2.8 µm) [10,18].

### 2.3. Crystallinity

The crystalline or amorphous nature of the therapeutic ingredient in the powder can influence its solubility, dissolution rate, and bioavailability in the lungs. Amorphous forms have a more uniform, smaller particle size, which improves aerosolization efficiency and lung penetration. They can reach smaller airways and alveoli more effectively, making them suitable for treating diseases like asthma or chronic obstructive pulmonary disease (COPD). They have a higher dissolution rate, allowing them to be absorbed more readily into the bloodstream. In contrast, crystalline forms are more prone to agglomeration, leading to uneven distribution in the lungs. The amorphous forms are thermodynamically unstable and tend to transform into a more stable crystalline form, thus losing their advantages and making it challenging to formulate them in a stable form [19,20].

### 2.4. Hygroscopicity

Hygroscopic powders can absorb moisture from the environment, potentially altering their size, flow properties, and stability. This can impact the consistency of aerosolization. This process affects particles’ properties, such as bulk density and aerodynamic size [21]. Spray-dried colistin powders showed significant moisture absorption of up to 30% in a 60% humidity environment, reducing the fine particle fraction from 80% to 63.2%. Powders stored at 90% humidity tended to agglomerate and were unable to aerosolize effectively because of the higher humidity in the environment [22]. Emery et al. reported a gradual decrease in the aerosolization of hydroxypropyl cellulose with increasing moisture content, while Respitose^®^ ML001 aerosolization remained stable under similar conditions [23].

### 2.5. Surface Charge

Surface charge, expressed as zeta potential, can influence particle aggregation, electrostatic interactions, and deposition patterns in the respiratory tract. Electrostatic charges on particle surfaces are influenced by factors such as particle size, surface characteristics, and surface energy. Larger particles have rougher surfaces, leading to structural disorder and minimal moisture absorption [24]. A study by Kaialy et al. on mannitol particles revealed that the net electrostatic charge increased with a decrease in particle size, which can be attributed to the large active surface area of smaller particles. Such enhancement in surface charge strengthens particle cohesion and adhesion to other particles and the inhaler device’s inner surfaces, resulting in a reduced fine particle fraction. Surface shape and morphology also significantly influence surface charge acquisition [25]. Spherical particles have lower charge acquisition tendencies than elongated particles, while rough-surfaced particles have a greater propensity for charge exchange due to increased contact areas between particles and device surfaces. Electrostatic charges also impact aerosolization during dry powder inhalation, as powder accumulates charge, which is then transferred to therapeutic ingredients. Optimizing surface charge during therapeutic ingredient formulation is crucial, as computational lung models explain particle deposition patterns in the airways. Electrostatic charges contribute to deposition through cohesive attraction, particularly in the lower airways [10,26].

## 3. Carriers and Excipients Used for Inhaled Dry Powder Formulations

As previously described, several physicochemical properties of the dry powders need to be optimized to achieve the desired therapeutic efficacy of the inhaled drugs. Additionally, these dry powder formulations should possess excellent physical and chemical stability, a high emitted dose, and satisfactory dose reproducibility. Micron-sized particles that are suitable for pulmonary delivery usually exhibit a cohesive and adhesive nature, leading to poor flowability and aerosolization performance. This can be overcome by the appropriate and balanced use of excipients [27]. Excipients, or carriers, are inert molecules that are intentionally added to APIs to enhance their stability and mechanical properties, such as flowability, which further positively influence their therapeutic efficacy. Lactose is one of the most commonly used excipients for the pulmonary delivery of dry powders. The Food and Drug Administration (FDA) has approved a limited number of excipients in its list of inactive materials. The safety of the excipients is crucial here. Any new excipient must undergo toxicity assessment according to the guidelines (ICH, M3, S3A, and S7A) [28]. The evolving pulmonary drug delivery systems demand the use of novel excipient molecules. Given the limited regulatory guidance and difficulties in toxicity assessment procedures, focus is being shifted towards natural and bioinspired excipients. Zillen et al. [27] extensively reviewed such excipients involving amino acids, sugars, lipids, and biodegradable polymers.

Considering the aforementioned concerns, recent research has focused on the development of DPI formulations without carriers or excipients while maintaining therapeutic efficacy. Evolution in particle engineering approaches has accelerated the development of such carrier-free formulations. Healy et al. [29] extensively reviewed lactose-free dry powder inhalation formulations, which involved novel particle engineering approaches such as coating by mechanofusion, spray drying, nano-in-micro formulations, liposomes, and several commercialized technologies such as PulmoSol™, Technosphere™, and PulmoSphere™. The upcoming section details such particle engineering approaches with a focus on the principles, working mechanisms, and innovations. 

## 4. Approaches for Particle Engineering

Particle engineering is pivotal in the development of inhalable therapeutic formulations, with two primary approaches: top-down and bottom-up methods. The top-down approach involves reducing the size of existing therapeutic ingredient particles through techniques like milling or micronization, making it cost-effective and conducive for therapeutic ingredients with known properties. However, it can pose challenges such as particle agglomeration and alterations in therapeutic ingredient characteristics, limiting its suitability for sensitive therapeutic ingredients. In contrast, the bottom-up approach focuses on creating nanoparticles or microparticles from raw materials or therapeutic ingredient molecules, offering precise control over particle size, shape, and distribution. This method can enhance therapeutic ingredient stability and bioavailability but is often more complex and expensive. The choice between these approaches depends on factors like therapeutic ingredient properties and therapeutic goals, and a combination of both methods may be employed to optimize inhalable therapeutic ingredient formulations, ensuring effective and safe therapeutic ingredient delivery.

To observe the research trends in particle engineering of inhaled therapeutics, the SCOPUS database (www.scopus.com, accessed on 27 September 2023) was searched with different keywords. Based on the different techniques reported, different keywords such as “Ball milling”, “Jet milling”, “High pressure homogenization”, “Spray drying”, “Freeze drying OR Lyophilization”, and “Spray freeze drying” in combination with “Dry powder inhaler OR Dry powder inhaler formulation OR DPI” were used. The year-wise publication trends in the past 36 years (1987–2023) were studied and estimated at 1251 publications in total. Figure 2 shows the trends in publications involving particle engineering approaches for the development of inhaled therapeutic particles. It can be observed that the milling approaches, though preferred and industrially well-implemented, attract less attention in research. On the other hand, spray drying is found to be highly explored for the development of inhalable therapeutic particles, perhaps due to its excellent scope for the development of particles with tunable properties. Furthermore, of late, spray freeze drying is picking up pace for the development of such dry powders for pulmonary delivery. In the upcoming sections, top-down and bottom-up approaches are discussed, explaining the fundamental principles, operations, and advantages and limitations of each. Additionally, with key examples, different aspects of each technique, such as optimization strategies, the effect of feed and processing parameters on product quality, and modifications to existing equipment setups, are discussed. Several other examples are mentioned in Table 1.

### 4.1. Top-Down Approaches

Top-down techniques for inhaled formulations involve designing and developing inhalable therapeutic ingredient formulations and delivery systems by starting with a macroscopic or bulk therapeutic ingredient material and then progressively reducing its size and modifying its properties to achieve the desired characteristics for effective inhalation. These techniques are often used in the pharmaceutical industry to create inhalable medications for the treatment of respiratory ailments such as asthma, COPD, and more. Top-down techniques are straightforward, repeatable, easy to prepare, and amenable to industrial production and reuse. However, the high cycle count needed to achieve the optimum medication particle size and the risk of aggregation over time lead to low physical stability [67]. This section presents important top-down techniques used for the preparation of inhalable formulations.

#### 4.1.1. Ball Milling

Ball milling is a size-reduction method for microparticle production. It consists of balls or rods made up of materials such as ceramic, agate, silicon nitride, sintered corundum, zirconia, chrome steel, CreNi steel, tungsten carbide, or plastic polyamide. The material is milled by a vessel’s rotation or vibration. During the movement, the balls collide with each other and the vessel’s inner wall. The balls’ impact and attrition reduce the medication particle size. The number of balls and quantity of starting material determine the vessel filling and milling intensity. Generally, the balls and starting material take up 50% and 25% of the jar, respectively, but the literature differs. A spinning jar rotates at 50–85% of the critical speed, where centrifugal force stops the balls from cascading. Attrition reduces particle size more than impaction and compression as the vessel’s spinning speed drops, resulting in finer particles but longer processing times. Rotation or vibration speed, ball size, density, and hardness affect particle size reduction. In addition, ball milling can molecularly blend amorphous therapeutic ingredient forms with hydrophilic excipients [68]. Recently, a wet ball milling approach with polar and non-polar solvents was used to develop an inhalable dry microparticle of amifostine as an alternative to its intravenous infusion. An optimized combination of amifostine (10 g), zirconia balls (50 g), and solvent (20 mL) was subjected to ball milling at 400 rpm for 15 min, followed by filtration and overnight drying at 60 °C. An in vitro assessment of aerodynamic properties was performed using a next-generation impactor (NGI), which revealed the superior performance of wet ball milling compared to jet milling. Furthermore, wet ball milling with non-polar solvents was found to be more effective, as they minimally affect the hydrate content and improve the therapeutic efficacy compared to polar solvents. Morphology, particle size, and FTIR analysis also concur with this conclusion [69]. Thus, though ball milling is effective for size reduction or amorphization, it is challenging to achieve effective control over particle properties and scale up the production of inhalable particle formulations. However, an interesting innovation for the development of inhalable formulations using a conventional milling approach is reported in a United States of America patent (US20110236492) [70] that possesses the potential to tailor the properties of particles. The inhalable formulation was developed by separately co-milling an API and carrier/excipient in the presence of an additive material. This is the key step: co-milling composite particles of an API and additive material or a carrier and additive material. This innovation allows the formation of powders with excellent powder properties and can be applied to any API. Furthermore, the separate co-milling approach also allows different additive materials with different quantities to be milled to obtain the desired properties of the formulation.

#### 4.1.2. Media Milling

Media milling is a more advanced form of ball milling, which involves a conventional wet milling operation to reduce the size of therapeutic ingredient particles in aqueous or non-aqueous liquid media. The liquid medium prevents therapeutic ingredient particles from adhering and compacting on the vessel wall and surfaces of the milling balls, increasing nanoparticle production. Media milling is a continuous process where the therapeutic ingredient suspension is circulated through the milling chamber, with the suspended particles separated from the milling medium by a screen before they leave the chamber (Figure 3a). Most reported experiments involve transforming the resulting therapeutic ingredient suspension into solid dosage forms like dry powders. The degradation of balls caused by vigorous mixing forces in the vessel is a significant disadvantage of media milling. Erosion residues from the milling medium may contaminate the finished product, causing chemical instability and potentially influencing product properties. Optimizing process variables, such as stirring speed and milling duration, can decrease erosion risk. The NanoCrystal^®^ technology from Elan Pharmaceutical Technologies is an example of media milling, and global regulatory organizations have approved the particle-engineered products [68].

#### 4.1.3. Jet Milling

Jet milling utilizes high-velocity jets of gas (usually air or nitrogen) to impact and disintegrate therapeutic ingredient particles. The particles are introduced into a milling chamber, where they collide with each other and the chamber walls, breaking down into smaller sizes. Jet milling, also known as fluid energy milling, is highly efficient, possesses the ability to produce particles in the submicron range, and is suitable for heat-sensitive materials. Particles are reduced from 20–100 mm to 10 mm by air jet milling. A rate-controllable feeder feeds materials into a chamber with high-velocity compressed air. Particles enter the air stream and strike the milling chamber wall at high speeds, causing impacts and attrition that break them down. Shear forces separate the particles, and a classifier removes the particles with a size below the predefined cut-off size. The centrifugal force from the fluted wheel’s high speed limits the particle size that can pass through the air exhaust outlet. In large tubular milling chambers, the presence of colliding air jets at the periphery and the exhaust of air at the center are responsible for generating centrifugal particle classification systems. The limitation of this approach is that it requires a longer time to produce finer granules. 

Overall, the method works for heat-sensitive therapeutic ingredients and meltable materials [71]. It can produce higher volumes of powder continuously, and several formulations are micronized using fluid energy milling to increase dissolution and solubility characteristics. Recently, for lung therapeutics, LTI-03 was developed; it is a peptide derived from caveolin-1-scaffolding protein for treating idiopathic pulmonary fibrosis. The developed peptide has therapeutic properties for preventing excessive growth and expansion of fibroblasts, which in turn restores lung balance and protects healthy lung epithelial cells. LTI-03 has been tested in murine models of fibrosis in both liquid and powder formulations [72]. An air-jet milling approach was used to successfully develop a formulation with an MMAD and FPF of 1.6 μm and 93%, respectively, when the aerosol was generated through monodose RS01 DPI [73,74]. A phase 1a clinical study is being conducted to assess the safety, tolerability, and pharmacokinetics of LTI-03 inhalation powder [74].

#### 4.1.4. High Pressure Homogenization (HPH)

HPH is a commercially scalable technique used for multiple FDA-approved medications, much like the wet milling technique [75]. Fluid energy milling may be required before homogenization to micronize starting materials, which decreases issues associated with homogenization gap clogging and milling time. An intensifier pump raises the pressure in a slurry feeding stream, which is typically made up of therapeutic ingredient coarse particles and stabilizer, to 100–2000 bars. When the suspension flows through the gap quickly, Bernoulli’s equation indicates that the liquid’s static pressure falls below its vapor pressure. The high-pressure stream subsequently travels through a relief valve, where an abrupt drop in pressure causes cavitation and strong shear force and impacts the size of the particle (Figure 3b) [76,77,78]. It provides scope for continuous processing and has minimal issues with equipment contamination. 

Dissocubes™ technology, owned by SkyePharma PLC, produces water-based nanosuspensions with well-defined particle sizes. This method has improved therapeutic ingredient bioavailability. In 1999, it was developed to incorporate Nanopure^®^ nanocrystals, which work well for hydrolysis-sensitive medications and thermolabile therapeutic ingredients. Supercritical CO_2_ has also been used for homogenization. Nanoedge™ technology combines high-pressure homogenization and precipitation, reducing precipitated particle size and size dispersion. Inhaled budesonide [79,80,81] and sulbutamol sulfate [82,83] nanoparticles have been produced using this method. Typically, HPH requires a shorter processing time than wet milling, ranging from less than 30 minutes to a few hours [83,84]. HPH is often coupled with other particle engineering approaches as an integrated pre-treatment step [67]. The homogenization pressure controls the flow, which further affects the processing time–temperature cycles, solid loading, and items to be loaded and processed. Additionally, these variables are mostly affected by the hardness of the particle.

### 4.2. Bottom-Up Approaches

Bottom-up approaches offer better control over particle attributes like size, shape, and crystallinity compared to top-down techniques. The bottom-up particle engineering methods for inhalable dry powders include spray drying, freeze drying, and spray freeze drying. These involve crystallization and solvent removal to create nanoparticles and can be categorized based on drying adjuvants into solvent evaporation techniques and antisolvent methods. Solvent evaporation produces dry powders that can be used directly or converted to nanosuspensions, which can further be aerosolized using nebulizers or pMDIs. Antisolvent techniques result in nanosuspensions in a liquid solvent/antisolvent combination. 

#### 4.2.1. Spray Drying (SD)

The most prevalent technique for producing powders with inhalable nanoparticles is spray drying. This technique involves atomizing a liquid feed solution containing the therapeutic ingredient into liquid droplets that come into contact with a stream of drying gas. When the solvent containing liquid droplets enters the SD chamber, the solvent evaporates and dry solid particles form (Figure 3c). The bulk of the dried product is recovered using a cyclone separator, and small quantities of the remaining fine powder are removed from the exit gas stream using filter bags or additional cyclones [85]. The usual topology of these spray-dried powders is hollow, wrinkled, or dimpled, which contributes to their lower density and makes them more suitable for inhalation [86]. 

SD has a unique advantage over other drying techniques in terms of processing conditions (feeding solution and instrumental conditions such as flow rate, temperature, etc.), and precise control of particle size can be performed. This advantage allows the manipulation of SD processing parameters that can affect the in vitro aerosol performance of inhalable powders [87]. In a study on the development of inhalable formulations of amphotericin B, the authors observed that geometric particle size (D_50_) was inversely proportional to the gas flow rate. Additionally, it was observed that higher inlet temperatures and lower aspiration rates resulted in higher product yields [88]. With such a design of experiments, the properties of the particles can be tuned as per the requirements. Another significant benefit of SD is its fast drying period, which might lessen problems with colloidal instability in particles. Even when working with expensive ingredients, SD is a scalable and continuous technique that is well-suited for the pre-clinical and clinical development of DPIs. It is important to note that heat-sensitive materials, such as heat-labile pharmaceuticals, biologics, and polymers having low melting points (polycaprolactone and D-tocopheryl polyethylene glycol 1000 succinate (TPGS), may be less suitable for spray drying [89]. 

Recently, SD has been successfully employed to prepare dry powders of small nucleic acids, such as small interfering RNA (siRNA) [55]. SD has also been utilized as a solidification technique to produce microparticles containing nanoparticle (NP) agglomerates for inhalation [9]. Additionally, SD has been used to produce inhalable microparticles for tuberculosis therapy, with different solvents affecting the properties of the resulting microparticles [90]. Furthermore, SD has been employed to prepare salbutamol-loaded albumin microspheres for targeted therapeutic ingredient delivery to the lungs [91].

SD offers excellent scope for modification to achieve particles with tunable properties. For instance, Lavanya et al. used a conventional spray dryer in which the atomization process was modified with the use of an ultrasonic nebulizer. The liquid feed was converted into aerosol and subjected to SD. With this, the developed powder showed spherical particles with porous surface morphology (Figure 4) and an MMAD ranging between 2.82 and 3.02 µm. The logic lies in the entrapment of air during the nebulization process, which escapes during drying, leaving the surface porous [92]. A similar atomization approach was used to improve the aerosolization performance of chromolyn sodium (CS), which is a commonly used therapeutic ingredient for the treatment of respiratory diseases. CS with three different solvent ratios showed 4 and 2.7 times higher fine particle fractions as compared to commercial and untreated samples. The particles obtained by the modified approach had a spherical shape, a higher FPF (57.68 ± 1.85%), and a lower MMAD (3.01 ± 0.22 µm) distribution compared to the SD samples. An in vitro study on the NGI apparatus revealed that the particles with ammonium bicarbonate showed depositions in stages 3 and 4. Despite excellent performance, limitations include unsuitability for temperature-sensitive ingredients, lower yield, and the development of a crystal structure, which further reduces the efficacy of the formulation [93].

In another work, PulmoSphere^TM^, a commercial product with higher porosity and lower tapped density, was developed using SD [29]. Small phospholipid-based PulmoSphere^TM^ compositions are porous particles with a 1–5 μm particle range and low tapping density [94,95]. PulmoSphere^TM^ uses perfluorooctyl bromide (PFOB) emulsion, the addition of distearoylphosphatidylcholine, and calcium chloride as a major endogenous pulmonary component surfactant. Their discontinuous-phase submicron droplets form emulsions. The immediate evaporation of PFOB from the emulsion during SD creates structural holes in the particles. Three different PulmoSphere^TM^ varieties are available, such as solution-based, suspension-based, and carrier-based, among which TOBI R^©^ Podhaler^TM^ (tobramycin) is a commercial product produced by solution-based PulmoSphere^TM^ [94,96]. 

Large porous particles are categorized by geometric sizes ranging from 5–30 μm [51,97]. In contrast to solid particles, large-sized porous particle formulations have excellent deep lung penetration and can bypass alveolar clearance macrophages [97,98,99]. An appropriate porogen (often ammonium bicarbonate) is usually used as a matrix. Due to the quick release of carbon dioxide and ammonia from ammonium bicarbonate, the engineered structure becomes permeable. Cyclodextrin, another typical porogen that can act as an osmogene, causes inner and outer osmotic pressure differences in the outer aqueous phases. During drying, the water enters the organic phase and creates a porous matrix. Recently, SD INBRIJA^TM^ (levodopa) inhalable large porous particles were approved by the EMA and FDA for commercialization as a replacement for dopamine as a fast reaction doubled with rapid growth plasma levodopa concentration for treating Parkinson disease [96,100].

In recent research, the thymic stromal lymphopoietin-neutralizing antibody fragment CSJ117 has been targeted to modulate asthmatic airway inflammation. PulmoSol^TM^ technology, which originated for SD insulin Exubera^®^ (Nektar Therapeutics, San Francisco, CA, USA), has not been used for CSJ117. A recent clinical trial (NCT04410523) found that moderate asthmatics showed promising results with a regular dosage of 4 mg for 12 weeks and decreased allergen-induced bronchoconstriction [74,101]. The potential new class of asthma treatments includes anti-TSLP medicines like CSJ117. Also, Shepard et al. [102] demonstrated the preparation of two different active drug molecules in a combined DPI containing small-molecule APIs and biotherapeutic molecules in single-unit operation. A lab-scale modified atomizer wand was designed for operating two different APIs, which require different spray solvents using two-fluid atomizers. The study demonstrated the combination of bevacizumab with erlotinib, cisplatin, or paclitaxel in a dry powder inhaler. All the SD products had an MMAD of 1 to 3 μm within the target range of lung delivery. These model systems were selected for local lung cancer therapy relevance. The resultant formulations maintained antibody biologic activity, attained target drug concentrations, and exhibited aerosol characteristics suited for pulmonary administration. Thus, this method showed advantages such as permitting combination therapy with one or more drugs for an API that is shear-sensitive and not suitable for milling; the necessity for a large dosage is incompatible with carrier-based DPI technology. Some APIs cannot be dissolved in a common volatile solvent to be spray dried.

One of the limitations observed for SD is the conversion of drug molecules from crystalline form to amorphous form, which might affect its therapeutic efficacy. BREATH LIMITED developed a patented technology (WO2011018531) [103] for the dry powder formulation of beclomethasone using spray drying and ultrasonication without the use of any excipient. The beclomethasone suspension was subjected to spray drying, which generated powder in amorphous form. This was later suspended in an anti-solvent and subjected to ultrasonication, which resulted in the formation of a dry powder formulation with a re-crystallized API.

##### Nanospray Drying

Traditional SD also has a very low yield due to the possibility of retaining a sizeable fraction (30–50% by weight) of dry powders in the cyclone or other spray dryer parts, with notably poor collection efficiency for particles smaller than 2 µm [9]. The development of nano-SD aims to overcome this restriction. This method makes use of a vibrating mesh to create droplets and collects tiny particles using electrostatic forces. In comparison to traditional spray drying, nano-SD improves sample recovery and yield and helps attain particles smaller than 2 µm and particles with better aerosol performance [104]. Additionally, nanospray dryers may be used to create nanoparticle-based powders directly, which can then be combined with different carrier molecules [105]. Bürki et al. [106] developed inhalable protein powders using trehalose as a stabilizer and β-galactosidase as a model protein using a nanospray dryer. The study reported that lower inlet temperatures, greater ethanol concentrations, and smaller spray caps increased product recovery by affecting enzyme activity. The protein was more stable when spray-dried without ethanol and with a bigger spray cap. Furthermore, Schoubben et al. [107] made capreomycin sulfate inhalable powders using the nanospray dryer B-90. Experimentally altering membrane pore size, inlet temperature, and solution concentration improved the process. Capreomycin particles synthesized with lactose had a 27% twin-stage impinger respirable percentage. Spray-dried capreomycin sulfate formulations were included in multi-drug-resistant TB clinical studies after these encouraging outcomes. L-leucine improves dry powder aerosolization in nanospray particle engineering for pulmonary medication administration.

#### 4.2.2. Freeze Drying (FD)/Lyophilization

FD is a multi-step process that begins with the freezing of suspensions containing cryoprotectants. Subsequently, the solvent undergoes sublimation under low pressure and temperature conditions in what is known as the primary drying phase. This is followed by a heating step to eliminate any residual solvent content, termed secondary drying (Figure 3d). Lyophilization is a very popular technique in the pharmaceutical industry, particularly due to its low-temperature operation, which allows for the preservation of the therapeutic ingredient’s stability and activity, making it suitable for heat-sensitive therapeutic ingredients. FD powders can accommodate a high therapeutic ingredient load, enabling the delivery of therapeutic doses in small inhalation volumes.

Unlike techniques such as SD and SFD, FD does not entail the atomization of the feed liquid into droplets. Consequently, achieving precise control over particle size in the resulting dry powders proves to be more challenging. In practice, FD of nanosuspensions often yields powders characterized by a diverse and uneven particle size distribution. This variation in particle sizes can lead to suboptimal in vitro aerosol performance and render them less suitable for inhalation purposes. Additionally, during the bulk freezing of suspensions, there is a propensity for irreversible aggregation of nanoparticles to occur. This phenomenon contributes to the lower aqueous dispersibility of dry powder formulations [108].

Furthermore, FD is burdened by the inherent drawback of extended processing times. This protracted processing duration makes FD a less favored choice in the realm of particle engineering for inhalable nanoparticle-based dry powders when compared to techniques such as SD and spray freeze drying. Consequently, pharmaceutical researchers and developers often opt for these latter methods, which offer greater control over particle size and morphology and are better suited for achieving the desired characteristics in inhalable therapeutic ingredient delivery systems.

#### 4.2.3. Spray Freeze Drying (SFD)

As the name suggests, SFD is a combination of both SD and FD. SD is highly popular due to several advantages; nevertheless, high-temperature operation brings about the thermal degradation of heat-labile active pharmaceutical ingredients (APIs). On the other hand, FD effectively protects the potency of APIs due to its low temperature, but a long drying time, high operational cost, and limited control over particle properties act as limitations. SFD offers the advantages of both SD and FD. In the SFD process, a liquid formulation containing the API and other excipients is atomized into fine droplets using a spray nozzle. These droplets are rapidly frozen by exposure to a cryogenic medium, typically liquid nitrogen (Figure 3e). Following freezing, the frozen droplets are subjected to freeze drying, during which the ice within the droplets is sublimated, leaving behind dry, porous particles. These particles often exhibit advantageous characteristics for inhalation, such as controlled particle size, improved dispersibility, and enhanced aerosolization properties. This instant freezing prevents the formation of large ice crystals and allows for the preservation of the API’s stability and bioactivity. The high-temperature effects of SD and the inability of FD to form uniform-sized particles can be overcome by SFD. The same processing and formulation parameters that are used for SD may also be used for SFD to regulate particle size. Additionally, the primary drying temperature during freeze drying was found to play a significant role. In a study on the SFD of voriconazole inhalable particles, the primary drying temperature was found to be inversely correlated with particle size. The design of the experiment also revealed that the primary drying temperature and voriconazole content had the most significant effect on the aerodynamic performance of the dry powder. Voriconazole content had a positive effect on the percentage of FPF due to its hydrophobic nature [109]. It is emphasized that SFD delivers much greater production yields than conventional SD techniques and is especially well-suited for heat-sensitive nanoparticles. The schematic representation of a typical SFD process is depicted in Figure 5. 

Studies comparing the SD and SFD methods have shown that the latter produces particles with improved in vitro aerosol performance [86,111,112]. The characteristic porous particles formed during the sublimation of water from frozen droplets in the SFD process are likely the cause of this advantage [54,86,111]. The porous structure reduces the density, making them lighter. Heavier particles tend to undergo collisions and deposition in the upper respiratory tract. In their work on the development of bromelain aerosols, Lavanya et al. used the SFD approach and maltodextrin as wall material. The porous structure of the microparticles resulted in a lower density in the range of 0.3–0.38 g/mL and an MMAD in the range of 2.97–3.08 µm, suggesting suitability for pulmonary delivery [113].

However, there is disagreement about whether SFD works better than SD in terms of these compositions’ ability to disperse in water [112]. SFD has been deemed preferable in some investigations; however, other studies have shown conflicting findings, attributing these variations to the production of ice crystals during freezing [112].

SFD has been used to prepare particles of poorly water-soluble therapeutic ingredients, such as ciclosporin, in combination with dissolution-enhanced carriers like mannitol [114]. SFD has also been employed to produce nanoparticle aggregates (or nano-aggregates) for inhaled delivery, with the choice of excipients and the nanoparticle-to-excipient ratio affecting the characteristics of the nano-aggregates [115]. SFD has been recognized as a technology that can achieve continuous pharmaceutical manufacturing and has been applied in the development of lyophilization as a downstream operation [110]. Overall, SFD offers a promising approach for the production of inhalable therapeutic ingredient formulations with improved aerodynamic properties, solubility, and therapeutic ingredient delivery capabilities [116]. Liao et al. (2019) reported the application of SFD to engineer the large porous particles of voriconazole for pulmonary aspergillosis treatment. In the dissolving test medium, a quick release of voriconazole from porous particles was observed. On the other hand, raw voriconazole took 2 h to fully dissolve. The SFD products have a less porous structure and are more hygroscopic than the SD particles [56]. 

SFD has also been reported to produce inhalable particles of antibodies. In a patented technology, an antibody or antibody derivative was used as an API, and α- or β-cyclodextrin or their derivatives were used as excipients. The feed solution was subjected to SD or SFD to develop inhalable particles (WO2022111547) [117]. SFD showed better aerodynamic performance compared to SD, particularly due to its lower density and porous particle nature. 

Though SFD-engineered particles are of uniform size and distribution with low bulk densities, SFD has drawbacks, such as reduced throughput brought on by the prolonged lyophilization procedure. Furthermore, the SFD method has substantial difficulties when used in mass production, which may account for why, despite its advantages, it has received less attention than SD in the manufacturing of inhalable nanoparticle-based powders.

It is important to note, however, that little is known about the effects of several processing variables, including primary drying temperature, and their interactions on the aqueous dispersibility of dry powders based on inhalable nanoparticles that have undergone spray freeze drying. To shed light on the creation of these mixtures, further research is required.

#### 4.2.4. Supercritical Fluid Technology (SCF)

SCF-CO_2_ has been used for controlled-release therapeutic preparations [8]. This method allows for the impregnation of bioactive molecules and amorphous polymers without the use of toxic organic solvents or elevated temperatures. It provides a clean, green, and effective alternative to traditional therapeutic ingredient-releasing and polymer processes, resulting in a high-purity product free of residual solvents [9]. On the other hand, there are other methods for therapeutic ingredient drying, such as using a supersonic spray dryer that enables rapid synthesis of submicron-sized active pharmaceutical ingredients (APIs) at room temperature [10]. Another method involves using a therapeutic ingredient dryer with a heating system, evacuating system, and blowing system to accelerate therapeutic ingredient drying and shorten drying time [11]. 

The use of SCF-assisted SD (SASD), which involves mixing and solubilizing carbon dioxide (CO_2_) in its SCF form from inside the feed as a continuous phase before spray drying, is an inventive way to overcome this restriction. With less energy input and drying temperatures that are generally below 60 °C, this method minimizes thermal stress on the nanoparticles [118]. In the study by Silva et al. [119], respirable microparticles were utilized to deliver therapeutic biomolecules deep into the lungs, achieved through functionalized gold nanoparticles. These gold nanoparticles were modified with biocompatible fluorescent coatings and peptides for lung cancer treatment. The researchers employed a method known as SASD (spray-assisted supercritical antisolvent drying) to incorporate functionalized gold nanoparticles into chitosan, creating nano-in-micros dry powders. What makes SASD particularly valuable is its rapid process that eliminates heat degradation, making it suitable for attaching medicinal molecules. Aerosol characterization tests demonstrated fine particle fractions (FPFs) in the range of 30–40% for customized formulations. These formulations exhibited superior biodegradation and release patterns, allowing for extended and controlled nanoparticle release and improved cellular uptake. Furthermore, the study involved encapsulating strawberry-like gold-coated magnetite nanocomposites and ibuprofen within a chitosan matrix using SASD, leading to excellent morphological and aerodynamic performance in dry powders with FPFs of 48–55%. Notably, ibuprofen was released rapidly at a pH relevant to lung cancer (pH 6.8), demonstrating the potential of SASD for creating dry gold nanocomposite powders for lung delivery of therapeutic agents. In another study conducted by Restani et al. [120], they developed dry inhalable nanoparticles based on POxylated polyurea dendrimers. Paclitaxel and doxorubicin-loaded nanoparticles were micronized with chitosan using SASD. Aerosol characterization tests indicated fine particle fractions of approximately 30% for the dry powder formulations. These engineered formulations exhibited a more potent chemotherapeutic effect than free paclitaxel when tested in adenocarcinoma cells in vitro, pointing towards the potential of inhalation chemotherapy.

SCF-CO_2_ possesses strong affinity and has excellent solvation power towards a variety of organic solvents, which are being used as anti-solvents for the development of engineered particles for inhalation. The precipitation of compressed CO_2_ antisolvent (PCA) and SCF anti-solvent process (SAS) approaches provide a significant benefit for obtaining particles of the required size [121,122]. Lin et al. successfully used the PCA approach to produce poly-L-lactide porous microparticles loaded with insulin. The resulting porous particles demonstrated 97% encapsulation efficiency and targeted aerodynamic deposition. The resultant particles were solvent-free residue, and thus low inflammatory reactions were verified under in vivo conditions, with a prolonged release pattern of insulin observed [123].

#### 4.2.5. Electrohydrodynamic Approaches

Electrohydrodynamic approaches, namely electrospraying and electrospinning, are widely popular for the development of nano-scaled fibers and particles, which are extensively explored for their encapsulation and delivery and several other applications. Typically, a natural or synthetic polymer solution is subjected to high-voltage direct or alternative current (1–40 kV) while being pumped through a blunt-ended stainless steel needle or capillary. Due to the high voltage, the accumulation of like charges at the tip of the needle takes place, which results in the conversion of a spherical droplet into a Taylor cone. Once the electrostatic repulsive forces overcome the surface tension, ejection of the jet of polymer solution takes place [124]. Furthermore, the transition of the Taylor cone into fibers (electrospinning) or particles (electrospraying) is governed by the product and process parameters, particularly the concentration of polymer [125]. The major advantage of electrohydrodynamic approaches is their non-thermal nature, which gives them the upper hand when compared to a well-established technique such as spray drying. Furthermore, electrohydrodynamic approaches hold the potential to meet the requirements of pulmonary drug delivery, such as precise control over particle size, narrow size distribution, absence of agglomeration, and high drug encapsulation efficiency [126].

For the treatment of non-small cell lung cancer, oridonin-loaded poly (D, L-lactic-co-glycolic) acid microspheres were developed using electrospraying. The developed particles have a rough, porous surface morphology suitable for pulmonary delivery. It was observed that the low voltage and longer distance between the needle and collection plate favor the formation of porous particles. Additionally, the inclusion of pore-forming agents such as NH_4_HCO_3_ is recommended. In vitro and in vivo characterization revealed that most of the drug was released within 20 hours of delivery, suggesting a high anti-cancer effect in rat models. In another study, ciprofloxacin was encapsulated in chitosan through a single-step electrospraying process for the treatment of respiratory infections. The use of chitosan as an excipient plays a major role here, as the positively charged polymer effectively interacts with mucosa and pathogenic bacteria. The characterization of prepared particles shows their compatibility with alveolar cell lines and antimicrobial activity against *S. aureus* and *P. aeruginosa* [127].

Lactose is a commonly used carrier for DPIs. Patil et al. used electrospray as a novel approach for engineering lactose particles for improved pulmonary drug delivery. Electrospraying has improved morphological and surface characteristics, resulting in better aerosolization properties compared to Respitose^®^ (SV003) [128]. Contrary to this application, electrospraying has also been reported to develop inhalable powders without the use of a carrier. In their work on celecoxib, Jahangiri et al. optimized the composition of the solvent mixture and drug concentration to obtain an MMAD of 2.82 µm, which is significantly lower than the untreated drug (4.73 µm). Furthermore, electrospraying was reported not to affect the drug’s crystallinity, functionality, or bioactivity [129].

Several drugs and bioactive drugs deliver synergistic benefits when delivered together. To develop such combinations of modifications, different modifications in formulations and equipment design can be made. One such approach is reported by Yaqoubi et al., where montelukast and budensonide were delivered simultaneously by the co-electrospraying approach for asthma treatment. The characterization of the developed co-encapsulates revealed that montelukast not only improved the aerosolization behavior of budensonide but also induced a synergistic pharmacological effect [130].

### 4.3. Hybrid Techniques

Preparation of engineered particles for inhaled therapeutic ingredient delivery is challenging, as the focus is on obtaining uniform particle size and high stability using a single preparation method. As a result, combination technologies composed of a pretreatment followed by high-energy milling technologies like high-pressure homogenization have also been developed [131]. Furthermore, the integration of both top-down and bottom-up approaches can be effective in the development of formulations with uniform and narrow size distributions with acceptable safety and efficiency. However, the industrialization of these combination approaches is constrained by their high production costs and challenging manufacturing procedures. By using a hybrid technology combining SD and HPH, Tao et al. (2016) developed resveratrol nanocrystal suspension with a significantly lower average particle size (192 nm) compared to HPH alone (569 nm) [132].

Recently, Carling and Brulls studied adaptive focused acoustic (AFA) milling. It and planetary bead mixing were used in conjunction with the wet milling of crystalline model medicines that have poor water solubility to obtain particle sizes suitable for sustained pulmonary administration. AFA milling emerged as a supplementary small-scale milling approach due to capabilities such as aseptic conditions, precise temperature control, minimal material loss, and concentration fluctuations. A size reduction up to an MMAD of 2 µm was observed for all model compounds within an hour. The milling efficiency, on the other hand, was heavily reliant on the compound qualities. Combining AFA milling with planetary bead milling provided 2–3 distinct monomodal particle sizes for all models. The dissolution kinetics of the model compounds’ varied particle sizes were observed and theoretically predicted, demonstrating that the dissolution kinetics may be defined, predicted, and considerably modified by changing the particle size [133].

In a recent study, Party et al. [134] demonstrated a hybrid technique that combines wet milling and SD of the poorly water-soluble API meloxicam (MX). Amorphization and an expanded surface area enhanced the MX’s ability to dissolve and diffuse and provide effective treatment for serious pulmonary diseases. The formulations displayed adequate aerodynamical characteristics, spherical particles with 3–4 μm in size, MMAD values of 1.5–2.4 μm, and FPF values of 72–76%. The accumulation in the deeper airways was shown under in vitro conditions by ACI and in the in silico measurements. 

A modified SFD with a microfluidic system (MF) was used to make SFD ciprofloxacin hydrochloride (CH)-embedded dry powders for inhalation. An active pharmaceutical ingredient-embedded DPI (AeDPI) is ideal for high-dose medication pulmonary administration. The SFD microparticle fine particle fraction (FPF) was affected by Leucine concentration and freezing temperatures. The optimized formulation was found to have a CH/Leu ratio of 7:3 and a freezing temperature of −40 °C. This combination showed minimally hygroscopic particles with prolonged storage stability, excellent therapeutic ingredient deposition, and remarkable aerodynamic performance. This study showed that MFSFD might replace the liquid nitrogen-aided approach in high-dose AeDPI investigations [135].

In another study, Baher et al. [136] created simvastatin nanoparticles using an emulsification and homogenization-extrusion technique and then spray dried them to reduce the size of the colloidal solution of mannitol-containing API to a respirable size. The in vitro deposition of the SD simvastatin formulation was performed using the NGI. Despite 60% of the medicine being deposited in the pre-separator, the lower stages of the NGI had enough simvastatin to distribute to the lungs. The nanoparticles’ FPF value of 19.75 ± 2.45% and MMAD of 1.33 ± 0.18 μm make them excellent for medication delivery to the lungs for the treatment of pulmonary arterial hypertension. 

## 5. Other Emerging Approaches in Inhaled Therapeutic Ingredient Preparation

Several novel technologies have recently emerged as promising methods for the preparation of improved DPIs. These innovative techniques represent exciting advancements in the preparation of dry powder formulations for pulmonary delivery, offering improved particle properties and potential benefits for therapeutic ingredient delivery [10].

One of these techniques is Particle Replication in Non-Wetting Templates (PRINT), which utilizes soft lithography. In PRINT, perfluoropolyether elastomers are used as molding templates on a silicone master plate to create micro- to nano-sized particulate matter with various shapes. Researchers have applied the PRINT technology used by Garcia et al. to prepare zanamivir-loaded microparticles for dry powder preparation. These microparticles exhibited significantly improved flow properties, with a 3.19-fold enhancement compared to conventional DPI technologies [137].

Another innovative method is inkjet printing (IJP), which offers precise control over particle morphology through a digital imaging system. In IJP, liquid materials are deposited dropwise onto suitable substrates to create particles with defined sizes and shapes. For instance, Lopez-Iglesias et al. employed IJP to develop salbutamol sulfate-loaded alginate aerogel microspheres, which showed a high porosity of 2.4 μm and enhanced FPF at 49.7% compared to powders produced using conventional methods [138]. This technology holds the potential for designing personalized aerosols with enhanced FPF and mass median aerodynamic diameter (MMAD) for powder delivery.

Thin-film freezing (TFF) is a rapid freezing technique for freezing an API and stabilizer solution under a fluid dynamic system. For rapid freezing, liquid formulations are quickly spread as a thin film on a cryogenically frozen surface, allowing the rapid conversion of liquid droplets into a solid mass, which can then be lyophilized to obtain dry powder. Researchers have reported that this method allows the formation of dry powders with smaller particle sizes, low bulk density, lower MMAD, respirable particle properties, and a higher delivery dose [139,140]. Recently, Pardeshi et al. [141] reviewed TFF-processed DPIs for the treatment of SARS-CoV-2 (COVID-19) using dry powder for inhalation of Remdesivir [142]. Using the TFF approach, a Remdesivir DPI was produced for the treatment of COVID-19 using Captisol^®^, mannitol, lactose, and leucine, resulting in adequate aerosol performance (93.0% FPF, <5 µm, and 0.82 µm MMAD) and one month of stability at 25 °C/60% RH. Amorphous Remdesivir has 20 times the solubility in simulated lung fluid using TFF-processed Remdesivir–Captisol^®^/lactose DPIs compared to crystalline Remdesivir–leucine/mannitol DPIs [143]. In another study, Sahakijpijarn et al. [139] used a tacrolimus (TAC) DPI, employing TFF technology to minimize lactose (LAC) for powder creation. 

Hot-melt extrusion (HME) is widely employed in the pharmaceutical industry to address challenges such as improving therapeutic ingredient solubility, masking the unacceptable taste, and formulating therapeutic ingredient products with extended-release. In HME, a viscous mass of therapeutic ingredients and polymers is prepared by heating them together above their glass transition temperature (Tg). This mass is then collected as a slug and micronized into a fine powder. Lin et al. applied HME to create inhalable itraconazole powder. Initially, itraconazole was jet-milled with mannitol (20:80 ratio), followed by extrusion through a twin-screw extruder, producing slugs subsequently processed through jet-milling to obtain inhalable-sized powder particles measuring 2.19 µm [144].

Nanotechnology and nano-scale delivery systems have always been successful approaches for the delivery of drugs and bioactive compounds due to their size, which offers a high surface-to-volume ratio. Nevertheless, in the case of pulmonary delivery, particles with a diameter of less than 0.5 µm are exhaled, creating difficulties in achieving therapeutic efficacy for nanoscale formulations. This has given rise to a new formulation involving nano-in-microparticle formulations, where a combination of various technologies is used to develop dry powders for pulmonary delivery with the benefits of nano-scale formulations. Such approaches involve Trojan Horse microparticles, liposomes, polymeric nanoparticles, and micelles and are extensively reviewed [11,145].

## 6. Clinical Trials for the Assessment of Inhaled Dry Powder Formulations

Dry powder for inhalation is becoming a popular system for local as well as systemic delivery of drugs. Inhalable particles are being developed through novel particle engineering approaches. The aerodynamic performance of these formulations is being assessed through several in vitro approaches. Nevertheless, human clinical trials are still required to assess the effectiveness of these formulations. In a clinical study, the safety and immunogenicity of the inhalable measles dry powder vaccine were tested on 60 adult males aged 18 to 45. The results revealed that the dry powder inhalable vaccine performed on par with the conventional subcutaneous vaccine. Furthermore, no adverse effects were observed in any of the subjects. The limitation of the study was that the trials were conducted on subjects with pre-existing measles antibodies [146]. 

Another clinical trial was aimed at assessing the safety of mannitol for systematic treatment of non-cystic fibrosis bronchiectasis. Patients with confirmed bronchiectasis aged 15 to 80 participated in a placebo-controlled, randomized, double-blind trial. The trial revealed that the patients treated with inhaled mannitol showed less airway mucus plugging on high-resolution computed tomography (HRCT) scans. The patients also reported higher compliance rates and tolerance [147]. During the COVID-19 pandemic, numerous drugs were tested for their clinical and therapeutic efficacy. This included corticosteroids [148], cytokines [149], antivirals, and vaccines.

## 7. Prospects of Dry Powder Formulations for Inhalation

The prospects for particle engineering technologies in inhalable particle delivery are promising and will continue to evolve as research and innovation advance. One key avenue of development is the refinement of particle design to enhance drug delivery precision. Advanced particle engineering techniques, including spray drying, supercritical fluid technology, and nanotechnology, enable the production of inhalable particles with tailored properties such as size, shape, and surface characteristics. These advancements can lead to improved targeting of specific regions within the respiratory system, optimizing the therapeutic effect, and reducing potential side effects.

Furthermore, there is a growing focus on the development of smart inhalable particle systems that respond to the patient’s needs in real-time. These systems can incorporate sensors and feedback mechanisms to adjust drug release or dosage based on individual patient responses and environmental conditions, improving treatment efficacy and patient outcomes. Additionally, biocompatible and biodegradable materials are being explored to create inhalable particles, minimizing concerns about long-term safety and potential toxicity. Such materials offer the potential for sustained drug release and controlled pharmacokinetics, which is particularly valuable for chronic diseases.

With the advent of personalized medicine, inhalable particle technology is expected to play a significant role in tailoring treatments to an individual’s genetic and physiological characteristics, enhancing therapeutic outcomes. This personalized approach, combined with advances in telemedicine and remote monitoring, can revolutionize the management of respiratory diseases and other conditions requiring inhalable drug delivery. The development of particle engineering technologies for inhalable particle delivery is marked by increased precision, patient-centric approaches, and the exploration of novel materials.

## 8. Conclusions

Pulmonary delivery of therapeutic agents and nutraceutical ingredients is becoming more popular than other administration routes. Of late, the field of pulmonary therapeutic delivery has witnessed remarkable progress, with a focus on developing innovative inhaled therapeutic ingredient products, particularly DPIs. The choice of particle engineering approaches, such as spray drying and jet milling, plays a pivotal role in optimizing therapeutic ingredient formulations. Additionally, the process parameters have a profound impact on the properties of the developed dry powders, which ultimately affect their potency, therapeutic efficacy, and suitability for pulmonary delivery. The application of novel hybrid techniques and precise control over particle properties can help in maximizing therapeutic efficiency. Additionally, inhaler devices themselves have a limited impact on medication absorption; thus, their correct usage remains crucial. Overcoming challenges related to particle deposition, humidity, mucus, anatomical constraints, and therapeutic ingredient bioavailability remains a complex endeavor. As research continues to dispel misconceptions and improve particle properties, including size, shape, density, surface charge, and moisture content, the field strives to enhance aerosolization, safety, and overall efficacy in pulmonary therapeutic ingredient delivery. This ongoing pursuit promises to bring about further advancements in respiratory therapeutics, ultimately benefiting patients and healthcare outcomes.

## Figures and Tables

**Figure 1 pharmaceutics-15-02706-f001:**
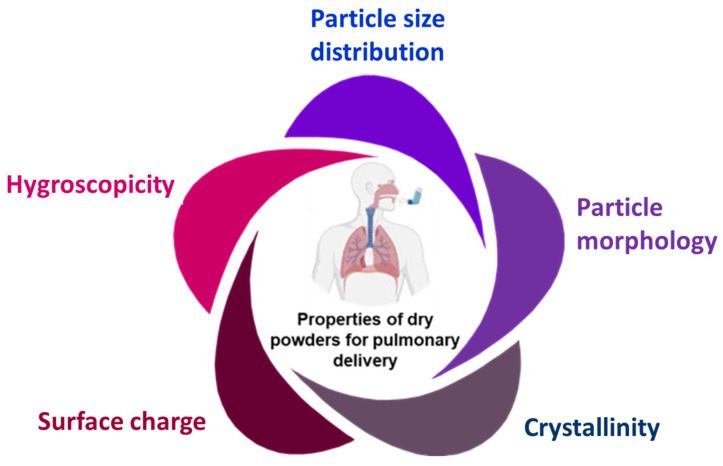
Major physicochemical properties of inhaled therapeutic ingredients.

**Figure 2 pharmaceutics-15-02706-f002:**
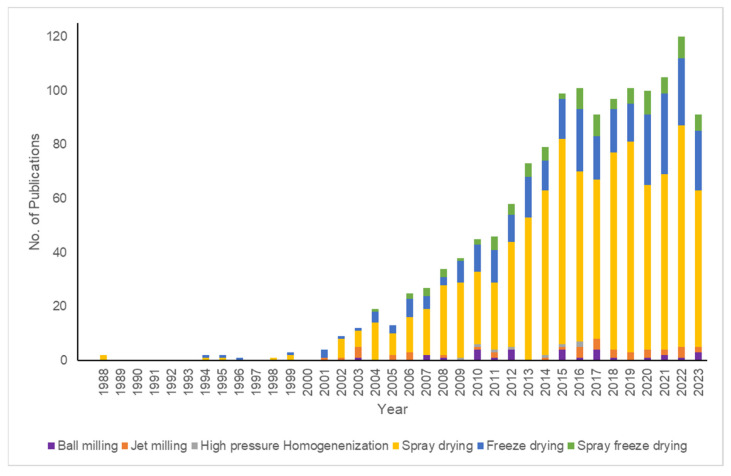
Year-wise trends in publications involving approaches to particle engineering during the past 35 years. Source: Online Scopus database (www.scopus.com) (accessed on 27 September 2023).

**Figure 3 pharmaceutics-15-02706-f003:**
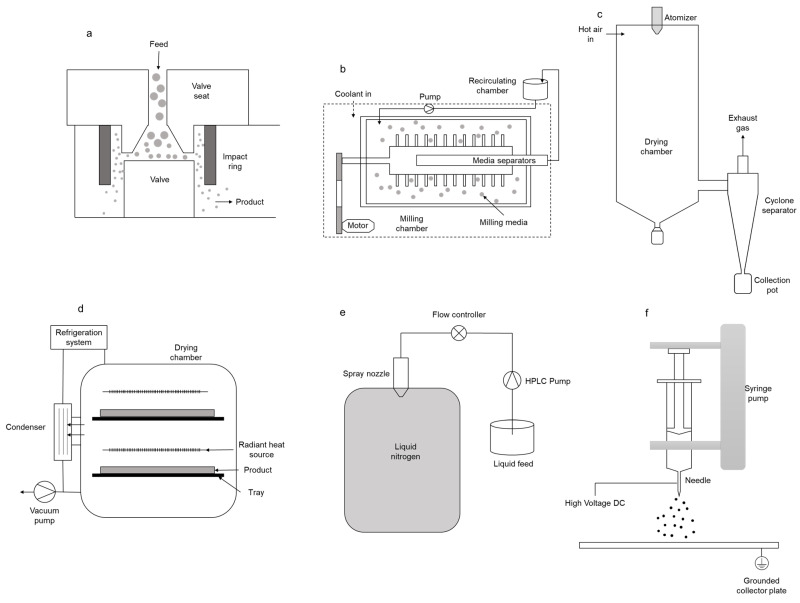
Schematic representation of particle engineering approaches: (**a**) media milling, (**b**) high pressure homogenization, (**c**) spray drying, (**d**) freeze drying, (**e**) spray freeze drying, and (**f**) electrospraying.

**Figure 4 pharmaceutics-15-02706-f004:**
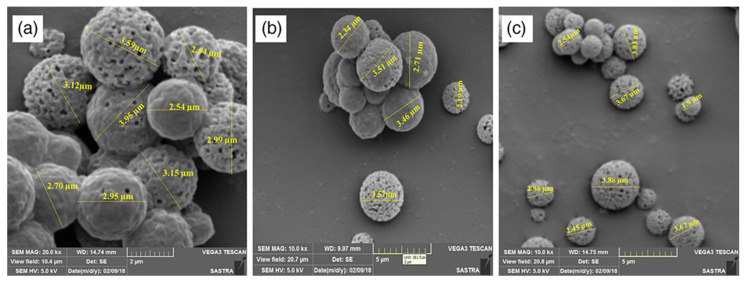
Morphology of the inhalable β-carotene microparticles prepared by modified spray drying with different core wall ratios: 1:10 (**a**), 1:25 (**b**), and 1:50 (**c**) (source: Lavanya et al. [29]).

**Figure 5 pharmaceutics-15-02706-f005:**
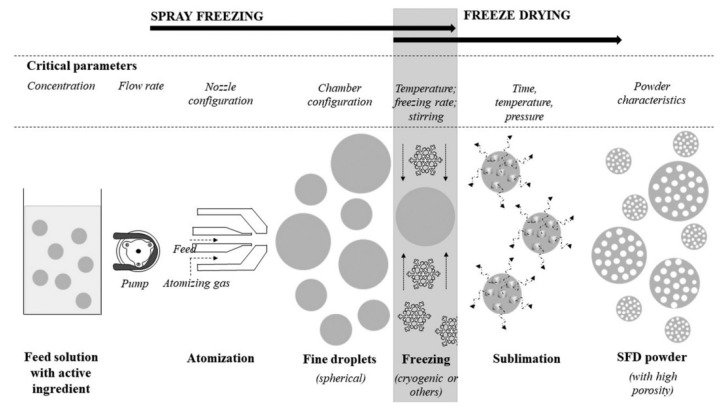
Representation of a spray freeze drying process (source: Vishali et al. [110]).

**Table 1 pharmaceutics-15-02706-t001:** Inhalable dry powders/aerosols prepared using different particle engineering techniques.

Technique	Therapeutic Ingredient/API	Additives	Particle Properties	References
Top-down approaches				
Jet milling and in situ micronization	Beclomethasone dipropionate	NA	~5 µm, FPF 40%	[30]
Micronization	Levodopa	L-Leucine	<5 µm Co-microionization with 2% leucine	[31]
Jet milling	Diclofenac	NA	2.36 µm, hollow crystal with different deposition patterns in NGI, jet-milled DF shows the best aerodynamic performance	[32]
Dry jet milling	Simvastatin	NA	2.2 µm, in vitro study with 60 L/min at MLI stage 3 filters with aerodynamic diameter < 6.8 µm	[33]
HPH	Itraconazole	Mannitol and sodium taurocholate	5.91 µm, with FPF 46.2 to 63.2% and increased solubility to 96 ng/mL	[34]
Combined wet milling with aerosol flow reactor	Indomethacin	Mannitol and L-leucine	0.96 µm	[35]
Single-step co-jet milling	Ciprofloxacin HCl and Colistin sulfate	NA	<5.4 µm, FPF 57.5 and 80.2% therapeutic ingredients, respectively	[36]
Bottom-up approaches				
SD	N-acetylcysteine	Soya phosphatidylcholine, Cholesterol, Polysorbate 80	7 µm MMAD, yield 71%, respirable fraction 30%	[37]
	Rifampicin	Soya phosphatidylcholine, Cholesterol, Hydrogenated soybean phosphatidylcholine	~2 µm, 70% loading of therapeutic ingredient, FPF 50%	[38]
	Rifapentine	NA	1.92 µm, FPF 83%	[39]
	Isoniazide	L-α-soybean phosphatidylcholine, Cholesterol, Mannitol	4.92 µm, FPF 15–35% with encapsulation of therapeutic ingredient 18–30%	[40]
Freeze-thaw followed by SD	Ciprofloxacine	Magnesium stearate and isoleucine	~1 µm, FPF 66–70%, encapsulation efficiency 79%	[41]
Co-SD	Docetaxel	Phosphatidylcholine, Cholesterol, Mannitol, Leucine	3.1 µm	[42]
	Moxifloxacin	Phosphatidylcholine, Cholesterol, Dextran	<5 µm, FPF 75% with deep lung deposition in rats	[43]
	Oseltamivir phosphate	Ovelecithin, Cholesterol, Leucine	~3.5 µm, FPF 35%, deposition studies show therapeutic ingredient release by twin-stage impinger	[44]
	Salmon calcitonin	Sodium tripolyphosphate, Chitosan, Mannitol	2.5–4.7 μm, FPF 63.5% with ACI	[45]
	Azethromycin	Not mentioned	1.6 µm	[46]
	Paclitaxel	Dipalmitoylphosphatidylcholine, dipalmitoylphosphatidylglycerol	2.3 µm, powder deposition in all stages of NGI, with a higher dose at the lower stages	[47]
	Tobramycin	Poly(lactic-co-glycolic acid), Poly(vinyl alcohol)	3.3 µm	[48]
	Tobramycin (PulmoSphere™)	Distearoylphosphatidlcholin, perflurooctyl bromide	~5 µm	[49]
	Zanamivir (Relenza^®^ Glaxo)	Mannitol, L-leucine, Poloxamer 188	2.3 µm, in vitro deposition of 58%, and 116% bioavailability relative to Relenza^®^	[50]
	Meloxicam	L-leucin, ammonium bicarbonate, sodium hyaluronate	In carrier-free formulations with 2.5 µm, the fine particle fraction and emitted fraction were higher for large porous particles than in non-porous formulations	[51]
	Dexamethasone palmitate (Pro-therapeutic ingredient of dexamethasone)	1,2-Dipalmitoyl-sn- Glycero-3- Phosphocholine (DPPC) and Hyaluronic Acid (HA)	Around 13 μm MMAD with a tap density of 0.05 g/cm^3^ and FPF of 40%. Large porous particles show sustained release, and the MMAD varies with therapeutic ingredient concentration.	[52]
SFD	Theophylline anhydrate and oxalic acid	NA	3.0 µm Geometric mean diameter of 7.20 µm	[53]
	Levofloxacin	Polycaprolactone, L-leucine, Mannitol	~4–5 µm	[54]
	Levofloxacin	Soybean lecithin, D-mannitol, L-leucine	5.6 µm	[54]
	Small interfering RNA	Mannitol	10–14.9 µm, an aerosol performance study using NGI showed an emitted fraction (EF) and FPF of 91% and 28%, respectively	[55]
	Voriconazole	Mannitol	3.8 µm, FPF 40% in NGI	[56]
	Octreotide acetate	Mannitol, ammonium carbonate	2.6 µm, FPF 40%, 88% bioavailability relative to commercial products	[57]
	Human IgG	Hydroxypropyl β-cyclodextrin, trehalose	~5.32 µm, FPF 51.29%, and particle behavior studied by the Anderson cascade impactor	[58]
	PlasmidDNA-Luc	Β-benzyl-L-aspartate N-carboxy-anhydride	7.6 µm, FPF 54%, in vitro inhalation study performed on ACI deposited in stage 3 and lower parts	[59]
SCF	Fluticasone-17-propionte	Poloxamer 188	~1.69 µm, FPF 61.9%, aerosol performance studied by NGI	[60]
	Salmon calcitonin	Inulin, Trehalose, Chitosan, Sodium taurocholate, β-cyclodextrin	2.2–2.9 µm, studies performed on Sprague-Dawley rats	[61]
	Ibuprofen	Chitosan	2.1–2.7 µm	[62]
	Plasmid DNA	Poly(D,L-lactic-co-glycolic) acid	Not mentioned	[63]
	siRNA	Chitosan	<10 µm	[64]
	5-fluorouracil	α-lactose monohydrate	Not mentioned	[65]
	Curcumin	Hydroxypropyl-β-cyclodextrin	~5.8 µm	[66]

NA: not available.

## Data Availability

No original data was reported (review article).

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
