# Peer review of "Engineering Inhalable Therapeutic Particles: Conventional and Emerging Approaches"

_pharmaceutics, 2023, doi:10.3390/pharmaceutics15122706_

Round 1

Reviewer 1 Report

Comments and Suggestions for Authors

The review covers the advanced development approaches for respiratory delivery, compared with past 100 years.

 The abstract describes the major content of the manuscript areas discussed. But, 3D printing approach not discussed in the main text.  

Add the references in first two paragraphs of the introduction. The references cited were more than 10 years old for some of the case studies and explanations. Better to reduce the ref numbers and cite the latest.

Write the mechanism of how particle technology differentiates in the physiological behavior of the system.

Include the future prospective of the technologies, that would be one of the key topic as per the journal submission guidelines.

Discuss patent technologies reported for the each of the discussed approach, if possible.

Author Response

Responses to the respected reviewer are enclosed herewith. Thank you.

Reviewer 2 Report

Comments and Suggestions for Authors

The submitted review article summarizes technologies to generate inhalable dry powder formulations. Great parts of the text are well readable and give a good introduction in available techniques for particle engineering for pulmonary delivery.

-The abstract has to be re-written because it is not consistent with the content of the review. There is no focus on personalized medicine or “synergistic ingredient delivery carriers”. I miss also the “smart devices for assisting health status monitoring and action”. The last sentence is an introductory phrase rather than a conclusion.

-In the keywords “Therapeutic efficacy” is mentioned but there is no mention of that in the review.

-The combination with the search term “Pulmonary delivery OR inhaled pharmaceuticals OR dry powder inhaler” presumably will identify also particles for pMDI. How did the authors exclude this?

-It should be explained, on which basis the examples in the text were selected. One option would be to list the APIs, which could be delivered with a specific technology. Especially for spray-drying, it is also suggested to describe the wide panel of materials used with this technique and describe the most important (for example closest to the market) formulations in more detail.

Minor

l.54: what means “higher scope” in this context?

Some abbreviations, especially the abbreviation of API, is introduced several times

Comments on the Quality of English Language

Few expressions are not commonly used. 

Author Response

(The authors gave the same response as above.)

Reviewer 3 Report

Comments and Suggestions for Authors

Overall, the review article looks interesting for the readers so in my opinion it is worthy of publication HOwever, there are few comments that need revision before publication. 

1. It is key to highlitgh and summarise which excipients are safe for lung admisnitration. Also, it is key to comment on those strategies that utilise nano-in -microparticles for lung delivery. See paper: Nebulised antibiotherapy: conventional versus nanotechnology-based approaches, is targeting at a nano scale a difficult subject?

2. Also, it it importan to highligth free-carrier formulations without lactose. See paper: Dry powders for oral inhalation free of lactose carrier particles

3. I think is important to include some graphics to summarise the important characteristics to take into account. 

4. Which critical factors tend to be included for each tehcnique when formulating dry powder inhalers? For example, in DOE studies whic variables need to be optimised. See paper: Targeting lung macrophages for fungal and parasitic pulmonary infections with innovative amphotericin B dry powder inhalers

5. It woudl be interesting to include a section about clinical trials going on as well as the scaling up of the formulations. 

Comments on the Quality of English Language

Some spelling errors and gramatical mistakes have to be checked.

Author Response

(The authors gave the same response as above.)

Round 2

Reviewer 2 Report

Comments and Suggestions for Authors

The authors gave a good explanation for the selection of examples in Table 1 in their reply and I suggest to include this also in the manuscript and to mention that the list is therefore not complete.

Author Response

Responses are enclosed herewith for kind consideration. Thank you. 

Reviewer 3 Report

Comments and Suggestions for Authors

Authors have addressed comments suggeste dby reviwers, so in my opinion the manuscritp is ready for submission. 

Comments on the Quality of English Language

mino spelling erros and grammar errors

Author Response

(The authors gave the same response as above.)
